# HSV-1 Infection of Epithelial Dendritic Cells Is a Critical Strategy for Interfering with Antiviral Immunity

**DOI:** 10.3390/v14051046

**Published:** 2022-05-14

**Authors:** Yang Gao, Jishuai Cheng, Xingli Xu, Xueqi Li, Jingjing Zhang, Danjing Ma, Guorun Jiang, Yun Liao, Shengtao Fan, Zhenye Niu, Rong Yue, Penglan Chang, Fengyuan Zeng, Suqin Duan, Ziyan Meng, Xiangxiong Xu, Xinghang Li, Dandan Li, Li Yu, Lifen Ping, Heng Zhao, Mingtian Guo, Lichun Wang, Yafang Wang, Ying Zhang, Qihan Li

**Affiliations:** Yunnan Key Laboratory of Vaccine Research and Development for Severe Infectious Diseases, Institute of Medical Biology, Chinese Academy of Medicine Sciences & Peking Union Medical College, Kunming 650000, China; xinglixu@imbcams.com.cn (X.X.); lxq12060806@163.com (X.L.); zhangjingjing940115@imbcams.com.cn (J.Z.); ynyxmdj@126.com (D.M.); jgr@imbcams.com.cn (G.J.); liaoyun@imbcams.com.cn (Y.L.); fst@imbcams.com.cn (S.F.); 13691256967@163.com (Z.N.); 13265173441@163.com (R.Y.); cploooo@163.com (P.C.); zengfengyuan0120@163.com (F.Z.); duansuqin@imbcams.com.cn (S.D.); mengzy0724@163.com (Z.M.); xiangxiongxu@mail.ynu.edu.cn (X.X.); lixinghangsir@163.com (X.L.); lidandan@imbcams.com.cn (D.L.); yuli@imbcams.com.cn (L.Y.); pinglifen0505@163.com (L.P.); zhaoheng@imbcams.com.cn (H.Z.); morrowfive@163.com (M.G.); wlc@imbcams.com.cn (L.W.); wang1706092017@163.com (Y.W.); cherryzhang629@126.com (Y.Z.); imbcams_qhli@163.com (Q.L.)

**Keywords:** herpes simplex virus type 1, herpesvirus entry mediator, dendritic cells, innate immune

## Abstract

Herpes simplex virus type 1 (HSV-1), an α subgroup member of the human herpesvirus family, infects cells via the binding of its various envelope glycoproteins to cellular membrane receptors, one of which is herpes virus entry mediator (HVEM), expressed on dendritic cells. Here, HVEM gene-deficient mice were used to investigate the immunologic effect elicited by the HSV-1 infection of dendritic cells. Dendritic cells expressing the surface marker CD11c showed an abnormal biological phenotype, including the altered transcription of various immune signaling molecules and inflammatory factors associated with innate immunity after viral replication. Furthermore, the viral infection of dendritic cells interfered with dendritic cell function in the lymph nodes, where these cells normally play roles in activating the T-cell response. Additionally, the mild clinicopathological manifestations observed during the acute phase of HSV-1 infection were associated with viral replication in dendritic cells.

## 1. Introduction

Herpes simplex virus type 1 (HSV-1), a member of the α-subgroup of the human herpesvirus family [1], is a virus that causes oral labialis, herpes progenitalis, and herpetic encephalitis in humans [2,3]. This viral pathogen has been found to spread widely among people of various ages and to cause infectious diseases that seriously impact the quality of life of individuals [4]. As an enveloped double-stranded linear DNA virus, HSV-1 has a genome approximately 150 kb in length and begins encoding following a linear time program under a tightly genomically regulated mechanism. The complicated genomic structure and transcriptional mechanism of HSV-1 allow it to express various structural and nonstructural proteins [5], including 12 glycoproteins (e.g., gD, gC, gE, and gG), which are expressed on the viral membrane and interact with different cellular receptors in various cells to achieve viral infection and spread in vivo [6]. Some virally encoded proteins can interact with molecules involved in the host immune system and inflammation and might interfere with their physiological roles in antiviral immunity [7]. HSV-1 infection leads to characteristic pathological lesions and an immune response [8]. The fact that HSV encodes a series of functional molecules to block or weaken innate and adaptive immunity has impeded the development of curative drugs and preventive vaccines [9]. Previous studies have suggested that the viral envelope glycoprotein gD interacts with HVEM, a member of the TNF-β receptor family expressed in some immune cells, including dendritic cells and T cells [10,11], which indicates that HSV-1 can actively enter dendritic cells unless it is captured by these cells following activation by signaling molecules associated with the innate immune response in infected tissue. During antiviral immunity elicited by viral infection, dendritic cells play important roles in capturing, processing, and presenting viral antigens, thereby activating the T-cell response [12]. During this process, dendritic cells dynamically mature from an undifferentiated state to a differentiated state as they receive antigenic stimuli, migrate to the lymph nodes, secrete various immunoregulatory molecules, and express distinct cellular surface markers [13,14]. All of these findings indicate that the role of dendritic cells in the innate immune response, based upon their intrinsic functions, might be affected by viral infection and intracellular replication, which might in turn affect the activation and production of adaptive immunity. Thus, more knowledge concerning the interaction between HSV-1 and dendritic cells during viral infection is needed to elucidate host pathology outcomes and immunologic mechanisms induced during HSV-1 infection. In the current study, HVEM gene-deficient mice were used, based on the observation of viral infections in mouse JAWSII-dendritic cells, to investigate the interaction between HSV-1 and dendritic cells and the impact of the viral infection of these cells on outcomes after pathologic infection and the immunologic responses involving innate immunity and specific antiviral immunity. The HSV-1 infection of epithelial dendritic cells can alter the dynamics of the innate immune response and local inflammatory activation, followed by activation of the characteristic adaptive immune response and clinical manifestations. The findings provide insight into host pathology and the immune response during HSV-1 infection.

## 2. Materials and Methods

### 2.1. Ethics Statement

C57BL/6 mice were purchased from Beijing Vital River Laboratory Animal Technology Co., Ltd. (Beijing, China). [animal license number: SCXK (Jing) 2016-0006], and HVEM gene knockout mice on the C57BL/6 background were constructed by Beijing Vital River Laboratory Animal Technology Co., Ltd. (Beijing, China). The experimental mice (half male and half female) were specific pathogen-free (SPF) grade and were aged 4–6 weeks. The mice were bred at our institute to expand the population and were housed in SPF-grade barrier facilities [laboratory license number: SYXK (Dian) K2014-0007]. The laboratory animals were cared for and used following the “3R” principle and animal welfare guidelines. The animal experiment process and animal-related care and welfare were reviewed and approved by the Animal Experiment Ethics Committee of the Institute of Medical Biology (IMB), Chinese Academy of Medical Sciences (CAMS) (approval number: DWSP 201803014).

### 2.2. Cell Lines

The African green monkey kidney cell line Vero (ATCC, Rockefeller, MD, USA) and mouse marrow immature dendritic cell line JAWSII (ATCC, Rockefeller, MD, USA) were purchased and maintained in the viral immunization room of IMB, CAMS. Vero cells were cultured with Dulbecco′s modified Eagle medium (DMEM; Corning, NY, USA) supplemented with 10% fetal bovine serum (FBS; HyClone, Logan, UT, USA), 100 U/mL penicillin and 100 µg/mL streptomycin (IMB, CAMS, Kunming, China) in an incubator at a constant temperature of 37 °C. JAWSII cells were cultured in minimum essential medium α (MEMα; Gibco, Shanghai, China) supplemented with 20% fetal bovine serum (FBS; HyClone, Logan, UT, USA) and 5 ng/mL recombinant mouse granulocyte macrophage colony-stimulating factor (rMuGM-CSF; MedChemExpress, Shanghai, China) in an incubator at a constant temperature of 37 °C with 5% CO_2_. After viral infection, the culture medium was changed to cell culture medium supplemented with 2% FBS.

### 2.3. Virus

The WT HSV-1 strain McKrae was maintained by IMB, CAMS. The titers of McKrae were determined by standard viral titration in Vero cells. Virus samples were serially diluted 10-fold in serum-free DMEM, and each dilution (100 µL per well) was added to 96-well plates with eight replicates, in which each well contained 100 µL of Vero cell suspension at a concentration of 5 × 10^5^ cells/mL. After the plate was incubated at 37 °C in 5% CO_2_ for 7 days, the cytopathic effect (CPE) was assessed under an inverted microscope (Nikon, Tokyo, Japan). All virus-related experiments were performed under biosafety level (BSL) 2 conditions.

### 2.4. Isolation of Dendritic Cells from Mouse Skin and Bone Marrow

The small pieces of skin were incubated in a digestion solution containing 5 mg/mL collagenase I (Sigma-Aldrich, St. Louis, MO, USA), 2.5 mg/mL trypsin (Thermo Fisher Scientific, Waltham, MA, USA), and 1 U/mL DNase I (Sigma-Aldrich) in Roswell Park Memorial Institute (RPMI) 1640 medium for 1.5 h at 37 °C with shaking at 200 rpm. The digested supernatant was filtered through a 70-μm cell strainer to obtain a single-cell suspension [15]. Furthermore, after removing both ends of the femur and tibia from the euthanized mice, the bone marrow was repeatedly flushed out into a Petri dish with PBS via a syringe until the bone was completely white. The bone marrow suspension was collected and filtered through a 70 μm cell strainer to remove debris and muscle tissue. The cells were washed with RPMI 1640 medium, red blood cell lysis buffer (Solarbio, Beijing, China) was added to remove the red blood cells, and the mouse bone marrow cells were resuspended in RPMI 1640 medium containing 10% FBS [16]. An EasySep™ Mouse Dendritic Cell Isolation Kit (StemCell Technologies Inc., Shanghai, China) was used to select dendritic cells using immunomagnetic particles. In brief, all cells (1 × 10^8^ cells/mL) were placed in a round-bottom tube, and then rat serum and selection cocktail were added to the tube. The mixture was incubated at room temperature for 5 min, and then Rapid Spheres™ were added to the sample. The tube was placed in the magnet and incubated at room temperature for 3 min, and dendritic cells were labeled with magnetic particles. The suspension was removed, and cells attached to the tube wall were pipetted and resuspended in RPMI 1640 medium with 2% FBS. The separated cells were plated and collected after infection with McKrae to construct a proliferation curve (multiplicity of infection (MOI) = 0.1) and determine the cytokine levels (MOI = 0.1). The specific primers used are listed in Appendix A.

### 2.5. Analysis of Virus Growth in Cells

A growth curve was generated to detect the replication characteristics of McKrae in JAWSII dendritic cell lines (MOI = 0.01) and dendritic cells from bone marrow and skin (MOI = 0.1). The untreated JAWSII dendritic cells were deemed the experimental group, and the two groups of JAWSII dendritic cells treated with rabbit anti-HSV1 gD antibody (Bioss, Beijing, China) or rabbit anti-TNFRSF14 antibody (Bioss, Beijing, China) were deemed the antibody blocking group. Dendritic cells were incubated with HSV-1 for 1 h at 37 °C. The cells were then washed to remove cell-free virus and resuspended in RPMI 1640 medium containing 2% FBS. Cultured viruses were collected, frozen and then thawed at 8, 16, 24, 32, 40, 48, 56, 64, 72, 80, 88 and 96 h, and the virus titer was determined in Vero cells.

### 2.6. Mouse Experiment Design

C57BL/6 mice were used as the control group, and HVEM gene knockout mice on the C57BL/6 background (HVEM^−/−^ mice) were used as the experimental group. Each group comprised 70 mice, and each mouse was intradermally infected with 2 × 10^5^ plaque-forming units (PFU) of McKrae. The mice were weighed every day, and the survival-to-mortality ratio was evaluated over a 15-day period. Three mice from each group were sacrificed 12, 24, 48, and 72 h and 5 and 7 days after infection. Skin and lymph node tissues were collected from the armpits and groin, and viral load, cytokine levels, pathology, and immunity were assessed. Serum samples were collected 28, 56, and 84 days after infection, and the spleen was collected for lymphocyte isolation 7, 28, 56, and 84 days after infection.

### 2.7. Immunofluorescence and Confocal Microscopy

Skin and lymph node tissues were collected from infected mice and immediately frozen in liquid nitrogen. According to the protocol, the tissue sections were embedded in optimal cutting temperature (OCT) compound (Tissue-Tek, Sakura, Torrance, CA, USA) and sectioned on a cryostat (CM1850, Leica, Wetzlar, Germany) at a thickness of 4 µm. The sections were fixed in 4% paraformaldehyde for 15 min at room temperature and then blocked with 5% bovine serum albumin (BSA). A rabbit anti-herpes simplex virus strain F (human) polyclonal antibody (Invitrogen, Thermo Fisher, Shanghai, China) and Alexa FluorTM 647-conjugated AffiniPure goat anti-rabbit IgG secondary antibody (Invitrogen, Carlsbad, CA, USA) were used to detect HSV-1 antigen, while a mouse anti-CD11c antibody (Abcam, Shanghai, China) and Alexa FluorTM 488-conjugated AffiniPure goat anti-mouse IgG secondary antibody (Invitrogen, Carlsbad, CA, USA) were used to detect dendritic cells. The cell nuclei were stained with DAPI. A confocal microscope (TCS SP2, Leica, Wetzlar, Germany) was used to visualize and analyze the fluorescence signals. The percentage of dendritic cells colocalized with HSV-1 out of all the dendritic cells was calculated based on the observation of 50 fields.

### 2.8. Quantification of Viral Load by qRT-PCR

An Axygen^®^ AxyPrep Body Fluid Virus DNA/RNA Miniprep Kit (Axygen Biosciences, Union City, CA, USA) was used to extract total DNA from tissue samples from the experimental mice. The primer pairs and probes used in this assay were designed to detect the gG sequences of the HSV-1 genome (forward primer: 5′-TCCTSGTTCCTMACKGCCTCCCC-3′, probe: 5′-FAM-CGTCTGGACCAACCGCCACACA-BHQ1-3′, reverse primer: 5′-GCAG/ideoxyl/CAYACGTAACGCACGCT-3′). The real-time PCR amplification conditions used were as follows: 95 °C for 30 s followed by 40 cycles of 95 °C for 5 s and 60 °C for 30 s. qRT-PCR was performed using a Takara Premix Ex Taq™ (probe qPCR) Kit (TaKaRa Bio, Dalian, China).

### 2.9. Cytokine Analysis

According to the manufacturer’s protocol, TRIzol-A+ reagent (TianGen, Beijing, China) was used to extract total RNA from the tissues of mice or dendritic cells collected at different time points after infection, and the One Step TB Green™ Prime Script™ PLUS RT-PCR Kit (TaKaRa Bio, Dalian, China) was used for amplification. Mouse glyceraldehyde-3-phosphate dehydrogenase (GAPDH) was the normalization control gene. The relative expression levels of inflammatory cytokines in mouse tissues were normalized to their levels in the blank control group by using the comparative Ct (ΔΔCt) method. The specific primer sets used are listed in Appendix A.

### 2.10. Histopathology

The skin tissues of euthanized mice were fixed in 4% paraformaldehyde, dehydrated, embedded, and then cut into 4-μm-thick sections for hematoxylin and eosin (HE) staining. Pathological changes were examined with an optical microscope (Leica, Wetzlar, Germany).

### 2.11. Neutralizing Antibody Detection

Serum samples were serially diluted 2-fold with serum-free DMEM and incubated with McKrae at 37 °C for 2 h. The mixture was then added to Vero cells seeded in a 96-well plate, and the cells were incubated at 37 °C and 5% CO_2_. The CPE was observed after 7 days to determine the neutralizing antibody titer in each serum sample.

### 2.12. Flow Cytometry Analysis

The spleen was collected under aseptic conditions and prepared as a single-cell suspension using a cell strainer, and peripheral blood mononuclear cells (PBMCs) were isolated from the spleen suspensions using lymphocyte isolation solution (Dakewe Biotech, Beijing, China) according to the manufacturer′s instructions. For analysis in a flow cytometer (LSRFortessa™, BD, Franklin, NJ, USA), PBMCs were stained with APC-CD83 (BioLegend, San Diego, CA, USA). FlowJo software was used to analyze the total number of lymphocytes.

### 2.13. ELISPOT Assay

The spleen was divided into PBMC suspensions as described above, and a mouse ELISPOT kit (MABTECH Inc., Cincinnati, OH, USA) was used to measure the interferon (IFN)-γ and interleukin (IL)-4 levels following the manufacturer′s protocol. Briefly, the positive control stimulant phytohemagglutinin (PHA) (5 µg/well) and a specific stimulant (95% pure peptide: gB498-505: SSIEFARL) (Sangon Biotech, Shanghai, China) were added to a 96-well plate precoated with IFN-γ and IL-4. Then, splenic lymphocytes were added to the plate and incubated at 37 °C for 12–48 h. After the incubation step, the cells and medium were removed to allow the spots to develop. An automated ELISPOT reader (CTL, Cleveland, OH, USA) was used to count the colored spots. Spot-forming cells (SFCs) were T cells that produced HSV-1-specific IFN-γ or IL-4.

### 2.14. Identification of the Effect of HVEM Deficiency

A MightyAmp Genotyping Kit (TaKaRa Bio, Dalian, China) was used to extract DNA from the mouse tail tip and determine the genotypes of the mice. Specific primers surrounding and inside the mutated regions of the HVEM gene were designed (upstream primer (Tnfrsf14-seqF): 5′-CTGACGTGGTGTCTGGGAAG-3′; internal primer at the gene deletion site (Tnfrsf14-seqR1): 5′-GCTGCCCAGACAGAGCTAAG-3′; downstream primer (Tnfrsf14-seqR2): 5′-CAAAGGCAGCTGGGCATATTR-3′), the genes were amplified by PCR using specific primers, and the samples were analyzed by agarose gel electrophoresis.

At the same time, the spleen lymphocytes of C57BL/6 mice and HVEM^−/−^ mice were lysed by RIPA buffer (Solarbio, Beijing, China) and then Western Blot was performed to verify HVEM deficiency. Protein lysates were separated using SDS-PAGE. Afterwards, proteins were transferred onto a nitrocellulose membrane by semi-dry transfer. After blocking the membrane in 5% skim milk solution (BioFroxx, Guangdong, China) for 2 h at RT, the membrane was incubated with rabbit anti-TNFRSF14 antibody (Bioss, Beijing, China) or β-actin mouse mAb (Cell signaling, Shanghai, China) overnight at 4 °C. The antibodies were detected via Image Quant and ECL using BeyoECL Moon Western blotting detection reagent (Beyotime, Shanghai, China) after the membrane was incubated with the HRP-labeled goat anti-rabbit IgG (H + L) (Beyotime, Shanghai, China). All antibodies are diluted in 5% skim milk solution.

### 2.15. Statistical Analysis

The experiments were performed in triplicate, and all the data are expressed as mean values with their standard errors. Significant differences between groups were analyzed by two-way ANOVA (GraphPad Prism8.0.2; GraphPad Software, San Diego, CA, USA), and *p* < 0.05 was considered statistically significant.

## 3. Results

### 3.1. HSV-1 Enters Dendritic Cells via gD Binding to HVEM and Replicates in the Cells

The specific interaction between the HSV gD protein and the HVEM receptor on the cellular surface initiates the membrane fusion reaction and transmits a signal to heterodimer gH/gL followed by possible viral infection of the cells [17]. Here, we detected viral infection in the mouse JAWSII dendritic cell line mediated by the interaction between gD and HVEM, in which viral infection was observed in the presence of a specific antibody against HVEM or gD or in the presence of medium only, with different proliferation rates (Figure 1A). The dynamic titration of viruses during infection suggested that HSV-1 presented a higher replication rate in untreated JAWSII dendritic cells than in cells treated with specific antibodies against HVEM or gD molecules (Figure 1A). The viral load detection of these infected cells indicated there were more copies of viral immediate–early and late genes in the experimental group than the antibody blocking group, which supported the above result (Figure 1B). Further detection of transcripts of some innate responsive genes during viral infection, such as the members of the IFN family, GM-CSF, TNF-α, TGF-β, IL-4, and IL-6, indicated that virion binding to HVEM was similar to specific antibody binding to it and could lead to a cellular response through some immune signal factors (Figure 1C). Previous data suggested that HVEM interacted by physiological ligands LIGHT or BTLA can induce a powerful pro-inflammatory reaction in immune cells [18], and that HSV gD is a dual antagonist by competitive displacement of BTLA and non-competitive blockade of the binding of LIGHT [19]. In this case, in the event that viruses or the antibody binding to HVEM lead to a pro-inflammatory reaction of dendritic cells, including a higher expression of the IFN-α, IFN-β, and IFN-γ in groups of virus infection and adding antibodies of HVEM than those in group of adding antibody of gD, it should be understandable during post infection, as while higher expression of GM-CSF, TNF-α, TGF-β, IL-4 and IL-6 was found at 48 h after infection (Figure 1C). The upregulation of some surface markers of dendritic cell maturation, such as CD83 and MHC-I [20,21,22], was also observed in virus-infected JAWSII-dendritic cells compared with antibody-blocking cells (Figure 1D). These results show that HSV-1 enables the infection of dendritic cells via the interaction between gD and HVEM, followed by viral replication. This event leads to an innate immune response of dendritic cells against the virus followed by their transfer of viral antigenic information to the adaptive immune system.

### 3.2. HSV-1 Enters Dendritic Cells in Epithelial Tissue via the Interaction between gD and HVEM and Replicates in the Cells

Previous in vivo studies have concluded that viral debris and antigen fragments from lysed HSV-1-infected epithelial cells are usually captured or engulfed by dendritic cells, which can automatically process and present these antigen epitopes to T cells as they migrate to the lymph nodes [22]. Some HSV-1-infected dendritic cells do not fully migrate to the lymph nodes or stay in local tissue, but how these infected cells impact the immune system is still unclear [23]. Based on the observation of HSV-1 infection in mouse JAWSII- dendritic cells above, HVEM-deficient (HVEM^−/−^) mice (genotyped and identified with specific antibody against the HVEM molecule; Western blot shown in Appendix A) and wild-type (WT) C57BL/6 mice were intradermally infected with 2 × 10^5^ PFU of HSV-1. Epithelial tissues from the infection site were collected from the mice of both groups 12, 24, 48, and 72 h after infection, and the localization of viral antigen (detected using anti-herpes simplex virus strain F (human) polyclonal antibody) in dendritic cells (detected using an antibody against CD11c) was assessed. Additionally, the viral load in the tissues was measured. The distribution of HSV-1 antigen in dendritic cells was different between tissues from HVEM^−/−^ mice and those from WT C57BL/6 mice, as determined by measuring the degree of colocalization between the antigen and the dendritic cell marker in 50 fields: The colocalization percentages were 4% and 3% in HVEM^−/−^ mice and 15% and 16% in WT C57BL/6 mice 24 and 48 h after infection, respectively (Figure 2A,B). The viral load in these samples was 100 times lower in HVEM^−/−^ mice than in WT C57BL/6 mice 48 h after infection (Figure 2C) and remained unchanged in C57BL/6 mice (Figure 2C). These data suggest that infection by HSV-1 of dendritic cells might be different during the early period in HVEM^−/−^ mice and WT C57BL/6 mice and imply the possibility that the infection by the wild-type strain of dendritic cells might weaken the responsive capacity of dendritic cells, triggering stress inflammation and the innate immunity in local tissue, which could permit viral proliferation in the cells. In WT C57BL/6 mice, the observation that a higher colocalization of the virus and dendritic cells at 24 h after infection was followed by a higher viral titer in tissue at 48 h after infection suggested that because of the capacity of the virus to directly infect dendritic cells in HVEM^−/−^ mice, the virus might be trapped in local epithelial tissue by the host-stress response, including inflammatory and innate immune responses triggered by dendritic cells and other immune cells residing in epithelial tissues. In this case, the viral antigens are actively captured by dendritic cells in response to viral infection in the epithelial tissue of HVEM^−/−^ mice. However, the virus can enter dendritic cells via the interaction between gD and HVEM and replicate in the cells; additionally, some debris of infected epithelial cells is captured with the dendritic cells in WT C57BL/6 mice. These events logically lead to a discrepancy in the pathological and immunological outcomes between both groups of mice in downstream processes.

### 3.3. HSV-1 Infection in Dendritic Cells Restrains the Activation of Innate Immunity and Inflammatory Reactions

In most cases, the replication of viral agents can destroy host cells as they infect epithelial tissue, causing pathologic lesions in tissues [24]. This process leads to the activation of the innate immune response and inflammatory reactions in tissues followed by the recruitment of immune cells, including dendritic cells, macrophages, and neutrophils through the gradient effect of various cytokines and chemokines secreted by infected epithelial cells [25,26]. This process also primarily determines the antiviral immunity of the host [27]. Because dendritic cells are infected by HSV-1 early, we hypothesized that the viral infection of these cells might alter the cellular program and the processing, transfer, and presentation of antigens to activate adaptive immunity. To test this hypothesis, dynamic alterations in the transcription levels of various innate immune regulators and inflammatory factors in samples of local infected tissue from C57BL/6 mice and HVEM^−/−^ mice were analyzed. The transcription level of IFN-γ secreted by local infected tissue, a critical immune regulator that exerts antiviral effects, was five times higher in tissues from HVEM^−/−^ mice than in those from C57BL/6 mice 48 h after infection, and the transcription level of IFN-α, a key indicator of innate immunity activation, was three times higher in HVEM^−/−^ mice than in C57BL/6 mice and stayed at a higher level in HVEM^−/−^ mice for a period of time (Figure 3A). Notably, the transcription levels of IFN-α in C57BL/6 mice were low during the acute phase of infection (Figure 3A). The expression of some inflammatory factors, including TNF-α, IL-6, and GM-CSF, were upregulated in HVEM^−/−^ mice (Figure 3B), and the expression of IL-12 and IL-23α, which stimulate the proliferation of Th1 and Th17 cells, showed a similar trend of upregulation in this group (Figure 3C). Conversely, the transcription levels of the chemokines CCL28 and CXCL12, which control homeostasis of the immune system and the strength of innate immunity, were higher in C57BL/6 mice than in HVEM^−/−^ mice (Figure 3D). These data suggested that the viral infection of dendritic cells in C57BL/6 mice led to the inhibition of crucial immune regulators and inflammatory factors required to activate immune responses in epithelial tissue while increasing the transcription of molecules that act in the negative feedback regulation that maintains the stability of the innate immune system, which is probably beneficial for viral proliferation in tissue, in comparison to what happened in HVEM^−/−^ mice. To verify these data, dendritic cells isolated from the epithelial tissues of C57BL/6 and HVEM^−/−^ mice were cultured and infected with HSV-1 in vitro. In this experiment, HSV-1 actively infected differentiated dendritic cells and replicated in these cells (Figure 3E and Appendix A), leading to early alterations in the biologic phenotype of these cells, including their expression of IFN-α, IFN-γ, TNF-α, IL-6, and GM-CSF (Figure 3F). These data support the results of the mouse experiments.

### 3.4. HSV-1 Infection in Dendritic Cells Leads to Alleviation of Infectious Manifestations in the Acute Phase

HSV-1 infection leads to oral or genital herpetic lesions in the skin followed by spontaneously recurrent latent viral neurological infection [28]. However, no obvious clinical manifestations are usually associated with the initial phase of infection [29], which is usually thought to be due to the alleviation of the innate immune response and inflammatory reactions in epithelial tissue by certain virus-encoded proteins [27,30,31]. Considering the differences in the transcription levels of innate immunity-related signaling molecules and inflammatory factors in epithelial tissue between infected HVEM^−/−^ mice and C57BL/6 mice, the clinicopathological process in both groups was evaluated. The results suggested that infectious inflammatory reactions, including capillary injection, the infiltration of inflammatory cells, and the degeneration of epithelial cells (Figure 4A), were most severe on days 4 to 6 after infection and were then rapidly alleviated in HVEM^−/−^ mice (Figure 4A, Table 1). No typical pathologic lesions were observed in the epithelial tissue of infected C57BL/6 mice, although a slight infiltration of inflammatory cells was observed during the same period (Figure 4B, Table 1). However, increased epithelial cell necrosis and severe infiltration of inflammatory cells into the dermis layer were found in C57BL/6 mice beginning on day 7 (Figure 4B, Table 1). Interestingly, the change in body weight was different between the groups of mice, the body weight of HVEM^−/−^ mice being reduced slightly on days 3 to 6 and increasing beginning on day 7 (Figure 4C), while the body weight of C57BL/6 mice showed a persistent decrease (Figure 4C). More HVEM^−/−^ mice than C57BL/6 mice died in the early stage (Figure 4D). Previous data suggested that physiological inflammatory reactions and innate immunity in local tissue allow a rapid response to viral infection in epithelial cells that primarily depends on the activation of dendritic cells and other inflammatory cells through immune signals secreted by infected epithelial cells [32,33]. This stress reaction could restrict viral infection in local tissue to further eliminate viruses, destroy cellular debris, and subsequently activate adaptive immunity via the presentation of antigens to T cells by dendritic cells and other cells [34]. Our work suggested that, as dendritic cells are infected by viruses, the innate immune response, which is associated with inflammatory reactions, can be interfered with and lead to attenuated clinical–pathological processes in the acute infection phase, providing an environment that allows viral replication and spread in tissues. However, viral infection in dendritic cells is alleviated in HVEM^−/−^ mice, which might lead to the real pathological phenotypes mediated by the inflammatory reaction in the acute phase of infection, which include severe pathological lesions in local tissue and a higher death rate in the period of acute infection.

### 3.5. HSV-1 Infection Interferes with the Roles of Dendritic Cells in the Lymph Nodes

Considering that dendritic cells are powerful antigen-presenting immune cells [35], we speculated that the biologic phenotype of dendritic cells infected with HSV-1 might impact T-cell activation via antigen presentation. The colocalization of viral antigen and the marker CD11c in lymph node tissues collected from infected HVEM^−/−^ mice and C57BL/6 mice at different time points was observed under a fluorescence microscope. The colocalization percentage of viral antigen and the dendritic cell marker in lymph node tissue was obviously higher after 6 days after infection in HVEM^−/−^ mice than in C57BL/6 mice (Figure 5A,B), in contrast to that observed in skin tissue (Figure 1A,B). An analysis of the viral load in the lymph nodes showed a tendency for the viral load to increase in HVEM^−/−^ mice but to remain unchanged in C57BL/6 mice (Figure 5C). These results suggest that viral antigen was transferred to the lymph nodes by activated dendritic cells in HVEM^−/−^ mice but not in C57BL/6 mice. This was because the immunological role of dendritic cells infected by the virus in C57BL/6 mice was altered to some extent, and their capacity to transfer antigen to lymph nodes was weakened, which led to a lower viral load in their lymph nodes. The transcription levels of various signaling molecules, including IKKβ, p38, STAT1, STAT3, and STAT6, showed a sharp increase from day 6 after infection in HVEM^−/−^ mice, compared with the rapid decrease at the same time in WT C57BL/6 mice (Figure 5D and Appendix A). A similar trend was found for transcription factors related to T- and B-cell proliferation in the lymph nodes, indicating markedly increased ROXγt, Foxp3, and T-bet since day 6 after infection in HVEM^−/−^ mice compared with WT C57BL/6 mice (Figure 5D and Appendix A). These signaling molecules and transcription factors have been found to increase the activation of specific antiviral immunity by participating in the signal transduction process and activating downstream gene expression, regulating the development and differentiation of T cells and B cells, adjusting the balance of the intracellular microenvironment and other effects [36,37,38]. All of these results imply that the canonical process by which dendritic cells activate adaptive immunity in lymph nodes went unimpeded in HVEM^−/−^ mice but was hindered in C57BL/6 mice. Additionally, the percentage of T cells expressing the surface marker CD83 in the lymph nodes of C57BL/6 mice was increased (Figure 5E). The extracellular structural domain of CD83 can restrain T-cell proliferation mediated by dendritic cells, although the mechanism is unclear [39,40,41]. These data also support our hypothesis that HSV-1 infection of dendritic cells may interfere with the ability of these cells to activate T cells, probably via antigen presentation and other mechanisms.

### 3.6. HSV-1 Infection of Dendritic Cells Followed by Suppression of Innate Immunity Leads to a Weakened Specific Antiviral Immune Response

A previous study demonstrated that indicators of the immune response, such as neutralizing antibodies and the specific cytotoxic T-lymphocyte response, are activated during specific immunity elicited by HSV-1 infection [33]; however, the presence of these indicators does not mean that immunity enables the elimination of viruses in vivo or defends against viral reinfection [42]. This is due to the interference effect of viral infection strategies on the immune system [7]. The current study suggests that HSV-1 infection of dendritic cells may exert these interference effects and alleviate the immunologic phenotype because the titer of neutralizing antibodies was higher, as determined by the neutralizing antibody assay. This high titer is associated with a stronger T-cell response and greater IL-4 specificity in the ELISPOT assay in infected HVEM^−/−^ mice than in C57BL/6 mice (Figure 6A–C). These results indicate that the infection and replication of HSV1 in dendritic cells lead to an alteration of cellular immunological roles, including antigenic treatment, transfer and presentation, and the alleviation of specific antiviral immune responses through some unknown mechanism.

## 4. Discussion

HSV-1, a DNA virus with a large genome encoding dozens of nonstructural proteins that interact with components of the innate and adaptive immune systems [43], causes a complicated pathologic process associated with an abnormal immune response [44]. Our understanding of this process is incomplete, likely because of the complicated mechanism by which this virus interacts with various host molecules, including components of signal pathways, transcriptional regulators, and effectors of the innate and adaptive immune systems [9,45], causing the infectious mechanism of HSV-1 to be distinct from that of other viruses [46]. Previously, HVEM was identified as being capable of interacting with some ligands to create the diverse sets of intrinsic and bidirectional signaling pathways controlling, enhancing, or inhibiting inflammatory responses through the activation of the NF-kB transcriptional system [12,47]. Interestingly, HSV gD was found as a competitor of BTLA to bind HVEM or acting as a non-competitive blockade of LIGHT [19,48]. These data suggested a hypothesis that the HSV-1 infection of resident dendritic cells in epithelial tissue via the binding of the viral glycoprotein gD to HVEM expressed on the cellular surface, may interfere with the local inflammatory response and innate and adaptive immune systems. Based on this hypothesis and the ability of dendritic cells to present antigens during the generation of specific antiviral immunity, our data firstly performed using mouse JSAWII dendritic cells showed that HSV-1 replication in dendritic cells modulates cellular phenotypes, including by changing the transcription levels of various immune regulating factors, such as interferons and chemokines. These findings, in which the HSV-1 infection group turns on the HVEM-LIGHT pathway and activates pro-inflammatory signaling, and because HSV-1 infection with the anti-gD group does not work in this way, are supportive to this hypothesis. Further, in HSV-1 infection with the anti-HVEM group, higher expressions of various pro-inflammatory factors, including members of the IFN family, suggest that the activating effect of HVEM by gD is similar to that by the specific antibody against HVEM. These results imply that the innate immune response and inflammatory reactions associated with epithelial dendritic cells are directly and/or indirectly altered, possibly leading to persistent viral proliferation in local tissue with less interference by the host and lesser inflammatory lesions in the acute phase of infection. This process may explain the characteristic pathologic process and attenuation of the immune response in HSV-1-infected patients in the clinic. Our observation of a severe inflammatory reaction in epithelial tissue during the acute phase, and the enhancement of antiviral immunity during the recovery phase in HVEM^−/−^ mice but not in C57BL/6 mice, suggests the later had a certain balance resulting from the interaction between viral infection and the host response. Although the pathological and clinical outcomes of HSV-1 infection in the acute phase are mild because of the viral takeover of the host innate immune response through the infection of dendritic cells, the final result is the widespread infection of the cell population with this virus. Thus, the virus is seemingly the winner over the immune system. In summary, our work reveals a mechanism by which HSV-1 restrains the host immune system through its infection of dendritic cells and the modification of their immune activity. This mechanism increases the likelihood that the virus will survive and spread in exchange for lesser pathologic lesions in the host.

## Figures and Tables

**Figure 1 viruses-14-01046-f001:**
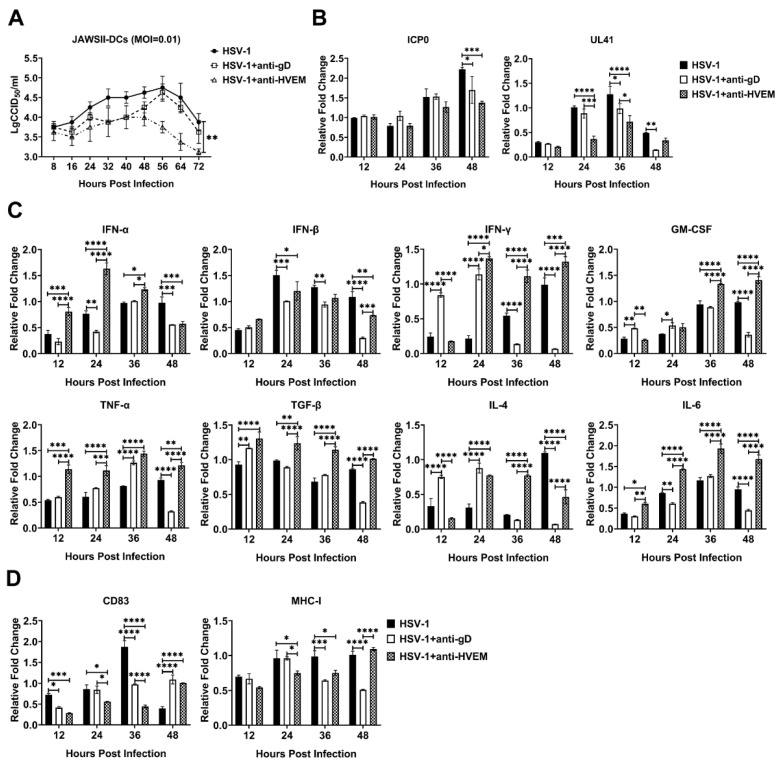
HSV-1 enters dendritic cells via the binding of gD protein to the HVEM receptor and replicates in the cells. The untreated JAWSII dendritic cells were deemed the experimental group, and the two groups of JAWSII dendritic cells treated with anti-gD specific antibody or anti-HVEM specific antibody were deemed the antibody blocking group. (**A**) Growth curves for the McKrae strains from the experimental and antibody blocking groups. (**B**) mRNA expression levels of genes and structural proteins such as ICP0 and UL41 of the McKrae strains in the experimental and antibody blocking groups. (**C**) Transcript levels of the IFN family, GM-CSF, TNF-α, TGF-β, IL-4, and IL-6 in the experimental and antibody blocking groups. (**D**) Expression of the maturation markers CD83 and MHC-I on the dendritic cell surface in the experimental and antibody blocking groups. The relative expression levels of inflammatory cytokines in JAWSII dendritic cells were normalized to their levels in the blank control group by using the comparative Ct (ΔΔCt) method. The data are from three independent experiments that were run in duplicate. Statistical significance was assessed by two-way ANOVA with Holm–Sidak adjustment for multiple comparisons (*, *p* < 0.05; **, *p* < 0.01; ***, *p* < 0.001; ****, *p* < 0.0001).

**Figure 2 viruses-14-01046-f002:**
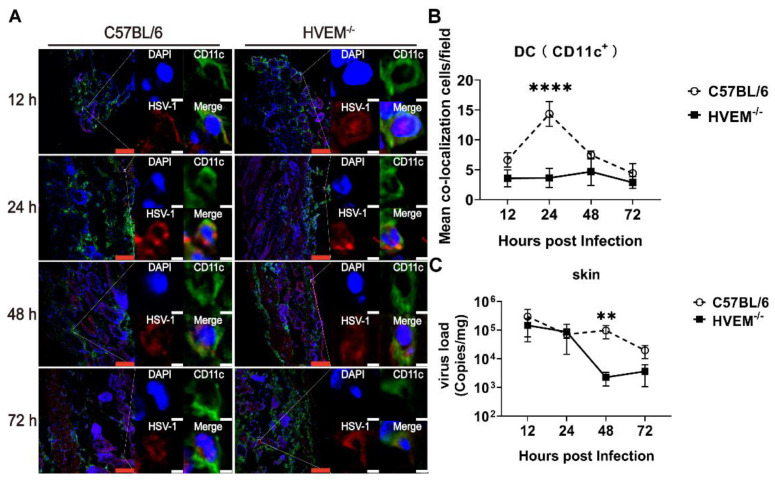
HSV-1 antigen enters dendritic cells in skin tissue and replicates after infection. (**A**) Representative confocal fluorescence images of HSV-1 expression (red), CD11c+ (green) and DAPI (blue) in the skin of C57BL/6 and HVEM^−/−^ mice 12, 24, 48, and 72 h after intradermal infection, taken at 20× magnification under a confocal microscope. The percentage of colocalized cells out of all the dendritic cells was based on the observation of 50 fields. The red scale bar is 100 µm, and the white scale bar is 5 µm. (**B**) Colocalization rate between HSV-1 antigen and a dendritic cell marker after infection (*n* = 3 per time point in each group). (**C**) Viral load in local skin tissues at 12, 24, 48, and 72 h after infection (*n* = 3 per time point in each group). Statistical significance was assessed by two-way ANOVA with Holm–Sidak adjustment for multiple comparisons (**, *p* < 0.01; ****, *p* < 0.0001).

**Figure 3 viruses-14-01046-f003:**
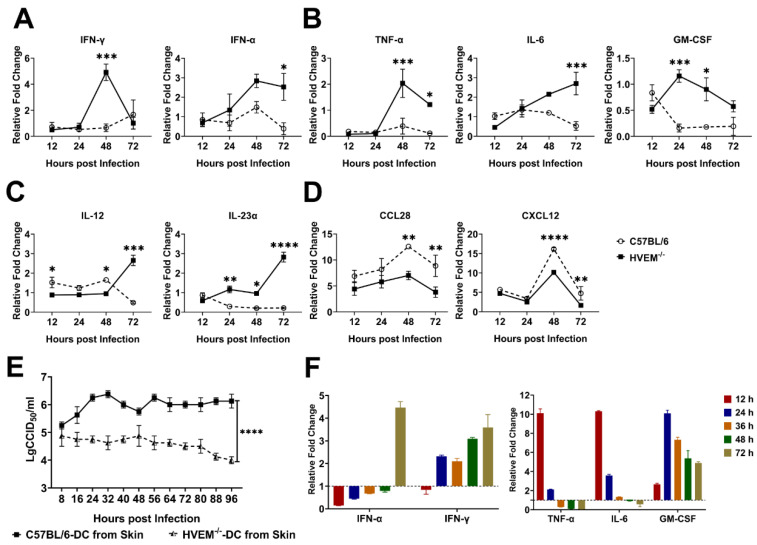
Change in the inflammatory response and the transcriptional expression of immune signaling molecules. (**A**) IFN-α, IFN-γ, (**B**) TNFα, IL-6, GM-CSF, (**C**) IL-12, IL-23α, (**D**) CCL12 and CXCL28 expression in the skin of C57BL/6 and HVEM^−/−^ mice after infection with HSV-1 was measured (*n* = 3 per timepoint in each group). (**E**) Analysis of the viral titer in dendritic cells from the skin and bone marrow of C57BL/6 and HVEM^−/−^ mice infected with HSV-1 and cultured in vitro for different times (*n* = 3 per timepoint in each group). (**F**) IFN-α, IFN-γ, TNFα, IL-6, and GM-CSF levels in mouse HSV-1-infected dendritic cells from the skin of C57BL/6 mice cultured in vitro. The gene expression of HSV-1-infected dendritic cells from C57BL/6 mice is shown as the fold-change (2^−∆∆Ct^) relative to the levels in samples from un-infected dendritic cells of C57BL/6 mice, which were used for calibration (*n* = 3 per time point in each group). Statistical significance was assessed by two-way ANOVA with Holm–Sidak adjustment for multiple comparisons (*, *p* < 0.05; **, *p* < 0.01; ***, *p* < 0.001; ****, *p* < 0.0001).

**Figure 4 viruses-14-01046-f004:**
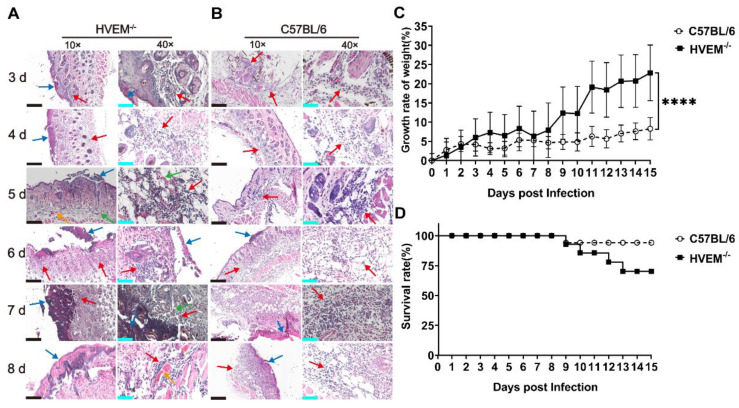
Clinical manifestations of HSV-1-infected C57BL/6 and HVEM^−/−^ mice. Pathological changes in the epithelial tissue of (**A**) HVEM^−/−^ and (**B**) C57BL/6 mice within 1–10 days of infection with HSV-1. Inflammatory cell infiltration and tissue necrosis are indicated with red and blue arrows, respectively, and tissue hyperemia and tissue bleeding are indicated with yellow and green arrows, respectively (*n* = 3 per time point in each group). The black scale bar is 200 µm, and the cyan scale bar is 50 µm. (**C**) Increase in weight and (**D**) the survival rate of C57BL/6 (circle) and HVEM^−/−^ (square) mice infected with HSV-1 (*n* = 12/group) during the 15-day observation period. Statistical significance was assessed by two-way ANOVA with Holm–Sidak adjustment for multiple comparisons (****, *p* < 0.0001).

**Figure 5 viruses-14-01046-f005:**
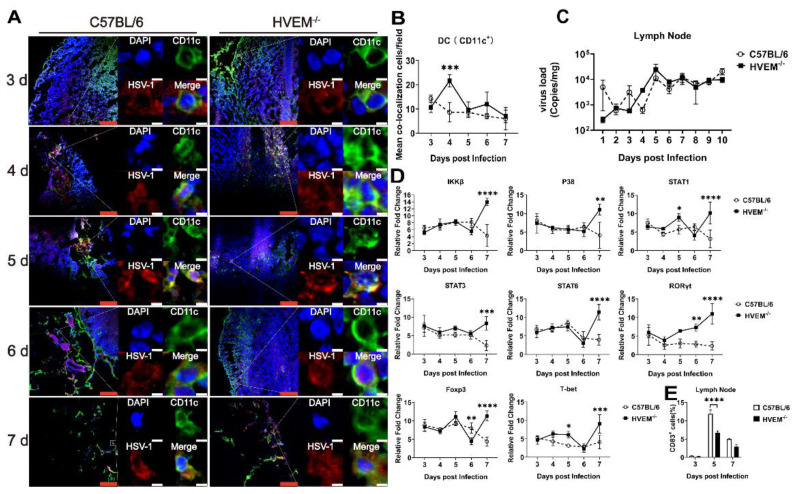
Effect of HSV-1 antigen on the activation of T cells after entering the lymph nodes. (**A**) Representative confocal fluorescence images of HSV-1 expression (red) and CD11c+ (green) in the lymph nodes of C57BL/6 and HVEM^−/−^ mice infected with HSV-1 antigen intracutaneously at 3, 4, 5, 6, and 7 days, taken at 20× magnification under a confocal microscope. The percentage of colocalized cells in the total dendritic cells was based on the observation of 50 fields. The red scale bar is 100 µm, and the white scale bar is 5 µm. (**B**) The colocalization rate of HSV-1 antigen and a dendritic cell marker after infection is shown (*n* = 3 per time point in each group). (**C**) Trend of the change in viral load in the lymph nodes within 1–10 days after infection (*n* = 3 per time point in each group). (**D**) Analysis of the transcription levels of various signaling molecules and transcription factors related to T-cell proliferation in the lymph nodes. The relative expression levels of inflammatory cytokines in mouse tissues were normalized to the level of the blank control group by using the comparative Ct (ΔΔCt) method (*n* = 3 per time point in each group). (**E**) Flow cytometry analysis and comparison of CD83^+^ cells in the lymph nodes within 3, 5, and 7 days after infection (*n* = 3 per time point in each group). Statistical significance was assessed by two-way ANOVA with Holm–Sidak adjustment for multiple comparisons (*, *p* < 0.05; **, *p* < 0.01; ***, *p* < 0.001; ****, *p* < 0.0001).

**Figure 6 viruses-14-01046-f006:**
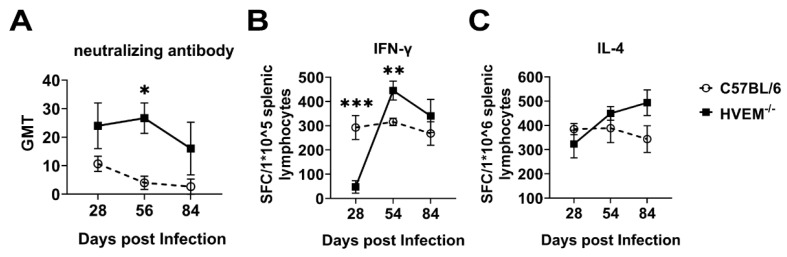
Dendritic cells infected with HSV-1 suppress innate immunity and weaken the adaptive immune response. (**A**) The levels of neutralizing antibodies in C57BL/6 and HVEM^−/−^ mice were measured at different time points (*n* = 3 per time point in each group). (**B**) ELISPOT analysis of gB_498-505_ elicited IFN-γ-secreting T-cell responses within the splenic lymphocyte population from C57BL/6 and HVEM^−/−^ mice. The samples were run in duplicate (*n* = 3 per time point in each group). (**C**) ELISPOT analysis of gB_498-505_ elicited IL-4-secreting T-cell responses within the splenic lymphocyte population from C57BL/6 and HVEM^−/−^ mice. The samples were run in duplicate (*n* = 3 per time point in each group). SFC, spot-forming cell. Statistical significance was assessed by two-way ANOVA with Holm–Sidak adjustment for multiple comparisons (*, *p* < 0.05; **, *p* < 0.01; ***, *p* < 0.001).

**Table 1 viruses-14-01046-t001:** Pathological analysis of C57BL/6 mice and HVEM^−/−^ mice infected with McKrae (*n* = 3 per time point in each group).

	C57BL/6	HVEM^−/−^
1 dpi	−/+	+
2 dpi	+	−/+
3 dpi	−	+
4 dpi	+	++
5 dpi	+	+++
6 dpi	++	++++
7 dpi	+++	++
8 dpi	+	+
9 dpi	−/+	−/+
10 dpi	−	+

Note: −, normal tissue; −/+, some proliferation of local lymphocytes; +, slight infiltration of inflammatory cells; ++, slight damage with inflammatory cell infiltration; +++, massive tissue necrosis with inflammatory cell infiltration and local vascular congestion; ++++, severe tissue necrosis with inflammatory cell infiltration and bleeding.

## Data Availability

Not applicable.

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
