# Peer review of "HSV-1 Infection of Epithelial Dendritic Cells Is a Critical Strategy for Interfering with Antiviral Immunity"

_viruses, 2022, doi:10.3390/v14051046_

Round 1

Reviewer 1 Report

Summary:

The authors aim to investigate the relationship between HSV-1 infection of dendritic cells and the host innate immune response. Authors report some interesting phenotype in HVEM knock-out mice infected with HSV-1, notably that HSV infection altered dendritic cell function such as T-cell activation by dendritic cell was diminished and limit activation of innate immunity. The study is interesting, but the authors are missing some necessary controls and need to address multiple discrepancies with the data presented. English editing is also needed.

Major comments:

  • The introduction contain is overall fine, but the writing makes it hard for the reader to digest. I would strongly advice performing some English/writing editing to make it easier on the readers. In particular, I would advise focusing on the multiple sentences that are too long by breaking them down or using more pronouns such as which etc. Secondly, in multiple occurrences the authors are a little too vague regarding specific mechanisms they introduced and should address these caveats by adding specific examples and references.

  • Line 231 to 233: Authors forgot to remove the description of the result section from the MDPI Viruses Word template.

  • Figure 1: Labels are unclear. From the results and methods, the authors block HSV-1 infection using either anti-gD or anti-HVEM/TNFRSF14 antibodies but labels show nomenclatures usually used for Knock-out gD-/- and HVEM-/-. Please correct the legend.

  • Figure 1C and 1D should include mock-infected control to show the baseline of these genes. In addition, it is very unclear what conclusion the authors are making from these. For instance, Interferon should be notably induced upon HSV-1 infection, however IFN-a/-b/-g are all over the place: increasing at times when blocking antibodies are used or decreasing at other time point. Authors should improve the description of their data and their relevance in the result section.

  • Figure 1D: Authors write that MHC-II is going up (line 250) with HSV infection. However, the graph presented in Fig 1D shows the inverse.

  • Figure 2: Authors should present a western blot showing that HVEM is actually knock-out in the mice.

  • Figure 2A/B: How was colocalization quantified exactly? Please update the method section. Out of the 4 time points tested (12h/ 24h/ 48h/ 72h) only one time point is statistically significant while colocalization does not really changes for the other time points. How do the authors explain that? There is the same observation for viral loads. In both cases the statistically significant point is way out the trend set by the other 3 points and the authors should explain that. Is there published studies that show these dynamics?

  • Figure 3F: I disagree with the statement (lines 341-344) that these molecules (IFN-α, IFN-γ, TNF-α, IL-6 and GM-CSF) are inhibited IFN-γ, GM-CSF are clearly upregulated, while TNF-α and IL-6 are strongly upregulated at early time points.

  • Figure 5: What is the viral titer in the lymph node?

  • How do the authors explained the discrepancies observed between skin tissues and lymph node regarding colocalization percentage of viral antigen and the dendritic cell marker (Figure 1A, B) and (Figure 5A, B) respectively?

  • The authors often use t-tests in panels where they have more than 2 conditions. This is not the correct statistical test. When they compare more than two conditions, they should use 1- or 2-way ANOVA followed by a post-hoc pairwise test corrected for multiple comparisons such as Tukey or Bonferroni tests.

Minor comments:

  • Line 39-40: Sentence “and begins encoding following a linear time program 39 under the tightly regulated mechanism of the genome” should be rephrased.

  • Sentence line 40 to 47: The sentence is unnecessarily long, and the authors should break it down to make it easier for the readers.

  • Line 48 to 52: Sentence should cite specific mechanism and should probably cite original articles rather than the authors own review.

  • Line 68: is there a dot in the middle of the sentence?

  • Line 107: I imagine the 5 in “105” should be in superscript.

  • Line 245-246: “Innate responsive genes” should be “innate immune response genes”.

  • Line 279: 2x105 should have the 5 as superscript.

Reviewer 2 Report

The study totled 'HSV-1 infection of epithelial dendritic cells is a critical strategy 2 to interfere with antiviral immunity, by Gao et al., is overall an interesting study. The authors have tried to shed more light on immunologic effect elicited by HSV-1 infection of dendritic cells by using HVEM-/- mice.

Some points of concern:

  1. In Figure 1B can the authors explain/discuss as to why the difference between the HVEM antibody blocked and unblocked condition is seen only after 48 hours after infection and not at early time points although ICP0 is an early gene of HSV-1.
  2. In figure 3F it is not clear if dendritic cells were used from control or HVEM-/- mice?
  3. The authors should discuss as to why more HVEM-/- mice than C57BL/6 mice died in the early stage in Figure 4D.
  4. The authors should discuss the result 6 in more detail.

Overall, i think they should highlight in the discussion section that studying HVEM-/- mice offers an interesting model system to study HSV-1 infection of dendritic cells.

Minor comments:

  1. Line 230-233 should be removed.
  2. Also, lines 238-243 have contradictory statements while the latter being correct. Authors should correct that.

Round 2

Reviewer 1 Report

Authors made some efforts to address my comments, but I still have an issue with two items:

  • Point 4: My point was not well addressed. Again IFN-a/-b/-g are all over the place: increasing at times when blocking antibodies are used or decreasing at other time point. This has not been explained well if at all but should be to some degree. Also why does the figure legend talk about mouse tissue when the work was done in cell line?
  • Point 6: Western blot need a loading control (such as GAPDH or B-actin)...
